# Revisiting Classical Controller Design and Tuning with Genetic Programming

**DOI:** 10.3390/s23249731

**Published:** 2023-12-09

**Authors:** Carlos A. García, Manel Velasco, Cecilio Angulo, Pau Marti, Antonio Camacho

**Affiliations:** 1Power and Control Electronics Systems, Universitat Politècnica de Catalunya, 08800 Vilanova i la Geltrú, Spain; andres.garcia.sanchez@upc.edu (C.A.G.); manel.velasco@upc.edu (M.V.); pau.marti@upc.edu (P.M.); antonio.camacho.santiago@upc.edu (A.C.); 2Intelligent Data Science and Artificial Intelligence, Universitat Politècnica de Catalunya, 08034 Barcelona, Spain

**Keywords:** genetic algorithm, genetic programming, control design, control tuning

## Abstract

This paper introduces the application of a genetic programming (GP)-based method for the automated design and tuning of process controllers, representing a noteworthy advancement in artificial intelligence (AI) within the realm of control engineering. In contrast to already existing work, our GP-based approach operates exclusively in the time domain, incorporating differential operations such as derivatives and integrals without necessitating intermediate inverse Laplace transformations. This unique feature not only simplifies the design process but also ensures the practical implementability of the generated controllers within physical systems. Notably, the GP’s functional set extends beyond basic arithmetic operators to include a rich repertoire of mathematical operations, encompassing trigonometric, exponential, and logarithmic functions. This broad set of operations enhances the flexibility and adaptability of the GP-based approach in controller design. To rigorously assess the efficacy of our GP-based approach, we conducted an extensive series of tests to determine its limits and capabilities. In summary, our research establishes the GP-based approach as a promising solution for automating the controller design process, offering a transformative tool to address a spectrum of control problems across various engineering applications.

## 1. Introduction

For nearly a century, control theory has played a significant role in enabling systems to operate in accordance with desired mathematical specifications [1,2]. Several control methodologies have been developed over the years, including PID control, adaptive control, robust control, stochastic optimal control, and others [3,4]. Each methodology features a unique regulator structure characterized by multiple parameters and components to be tuned. The variety of alternatives provided by classical control theory has enabled researchers to successfully control a diversity of systems and processes. However, the rapid technological advancements in society have given rise to increasingly complex systems and control challenges, necessitating even more advanced techniques to achieve effective control.

Artificial intelligence (AI) has emerged in the context of modern control as a critical discipline with a significant impact on both research and engineering applications. AI helps in the generation of “intelligent agents” capable of performing tasks or solving problems that are beyond the scope of conventional methods. Initially, AI applications were limited to advanced search engines, pattern recognition, speech recognition, data classification, data segmentation, and intelligent clustering [5,6]. However, as time passed, numerous disciplines recognized the potential of this technology and embarked on exploring ways to integrate it into their respective domains of expertise [7]. This trend has led to the emergence of application fields such as autonomous vehicles, motion biomimicry, self-learning for decision making, perception enhancement, object manipulation, social intelligence, and more recently, the fields of control and automation, to mention only a few [8,9,10,11].

The quick acceptance of AI in the control theory research domain in the form of intelligent control techniques [12] is mainly the result of machine learning (ML) progress, through strategies such as supervised learning (SL) and reinforcement learning (RL). In the early stages of investigation, AI technology was primarily integrated into control applications as a fault and anomaly detector or as an intelligent scheduler for module coordination [13,14,15,16]. However, the integration of ML methodologies rapidly escalated due to their efficient operation and quick responsiveness. The initial steps in this direction took form in examples such as [17,18], where the coordinated operation of a group of traffic lights was the result of an RL structure. Later on, investigations such as [19,20] began to describe complex systems, such as robots being controlled using SL or RL. From the lower levels of control to higher management levels, ML alternatives presented a reliable performance and optimal decision-making capabilities.

ML-based control offers solution to problems that conventional methods would not be able to solve. However, it is characterized by not resulting in an explicit analytical mathematical expression that can be readily implemented or modified using specific guidelines according to the system. Instead, control structures resulting from ML-based design techniques consist of networked layers of perceptrons, which are simplified mathematical representations of neurons, whose outputs activate according to the value resulting from the addition of their weighted inputs. In such structures, the human involvement is limited to participating by determining how many elements there are and what types they are comprised of. The values of the structure’s weights are automatically adjusted by the computer during the AI training process. Consequently, ML-based controllers can be perceived as functional “black boxes” that accomplish the specified goal, but their decision-making logic remains unknown to developers. For most of the cases, this limitation might be irrelevant, as the main goal is merely to solve the control problem. However, when the goal includes (i) understanding how to solve the problem through analyzing the controller equation and (ii) understanding the controller’s limitations and stability properties prior to its deployment, ML algorithms may not be suitable for development.

Machine learning (ML) has an older alternative that can provide solutions to control problems while generating structures that researchers can understand, analyze, and further improve using their knowledge, if necessary. Evolutionary techniques have been part of the AI research community since the early 1980s [21,22], and over the years the techniques have been successfully implemented in a number of control research projects [23,24,25,26]. The main evolutionary techniques in this AI branch correspond to genetic algorithms (GAs) [27] and genetic programming (GP) [28,29]. In the domain of evolutionary techniques, GAs identify and optimize parameters of an input–output system without varying their lengths and compositions. Conversely, GP is a methodology used to optimize both the structure and parameters of the system. In this case, the size and shape of the controllers dynamically change during the evolution process until an optimal structure that fits the specified requirements is found. Consequently, if properly implemented, GP allows for the generation of mathematical equations that are able to solve control problems, even complex ones.

When performing automatic controller design and tuning using genetic programming (GP), many factors influence the algorithm’s output, that is, the controller structure and its parameters. Certain aspects of this technique become more relevant to the control engineering side, while others seem to naturally align with the domain of optimization. Key control factors that strongly affect the outcome of the generation process are (i) the set of available functions and operators, (ii) the evaluation process, and (iii) the occurrence probability values assigned to the genetic functions. Therefore, within the context of GP-based automatic controller design and tuning, the contribution of this paper primarily focuses on the set of functions (also referred to as operators) and the fitness evaluation (function, process, and constraints) in order to show that GP is able to recreate classical controllers as an optimal solution for general processes.

In particular, for linear time-invariant plants, genetic programming suggests that, in the framework of the classical tracking problem, the most effective controller is a straightforward integral controller, which is exactly the controller that would be also suggested by classical control theory. The mere generation of an integral controller is not the primary contribution. The contribution lies in determining how to embed into a GP framework key aspects for proper controller synthesis, such as including time-domain operators like integrals and derivatives, or specifying objective functions accounting for the controller performance, robustness, and implementation feasibility. Such a GP-based approach becomes a promising technique for addressing intricate control problems, as far as it is able to mimic well-known solutions for simpler problems, and, unlike classical control, the procedure is general enough to be readily extended to more complex scenarios.

The remainder of this paper is structured as follows: Section 2 offers a concise introduction to evolutionary techniques; Section 3 provides an overview of the current state of utilizing genetic programming in the realm of automatic control; Section 4 details the proposed genetic programming approach; Section 5 and Section 6 shows implementation of selected test cases where the results are presented and discussed; and lastly, Section 7 offers concluding remarks and outlines directions for future research.

## 2. Evolutionary Techniques

Genetic algorithms (GAs) and genetic programming (GP) both operate on the principle of propagating generations of individuals by selection based on the “survival of the fitness” criterion. The individuals that are part of a generation are initially generated in a random way, and each one is evaluated and ranked using a fitness function related to the process goal. An individual in a GA corresponds to a set of values in a parameterized model to be optimized. In contrast, an individual in GP is a tree-based structure. As depicted in Figure 1, each tree is a combination of several nodes and branches. Usually, variables and numerical constants are located at the leaf nodes of the tree, known as terminals, while mathematical operators, which are located in the interior nodes, compute the values of nodes and leaves attached under it, commonly referred to as functions. Terminals are selected from a "terminals set", T∈T, and functions are selected from a "functions set", F∈F.

After the initial generation is populated with individuals, their parameters, known as genomes, are evaluated, and a fitness is assigned to them based on their performance on a cost function metric. Individuals with a higher fitness and ranking are more likely to progress to the next generation. There exists a set of usual genetic operations, also known as rules, that determine how individuals can be considered successful and evolve to the next generation [30]: elitism, replication, crossover, and mutation.

Successful individuals from each generation evolve to the next generation by means of any of the former genetic operations. However, a good practice includes newly generated individuals from scratch to promote diversity [31]. The evolution of generations continues until the performance of the cost function converges to a desired stopping criterion. As usual in machine learning algorithms, it is not guaranteed to converge to a global minimum. Nevertheless, these algorithms have achieved success in many applications, thanks to the alternative local minima that are returned at the end of evolution. Improvement of the evolution performance can be obtained by adjusting the number of individuals through generations and the occurrence probability of the genetic operations, guided by custom functions [32,33].

## 3. State of the Art

In this section, an overview is provided about existing work related to genetic programming (GP) for control theory, focusing on the challenge of automatic controller design and tuning. This is a research domain treated in the literature for more than 30 years, with favorable results. One of the pioneering works in this topic is [28], where an innovative method to solve problems in computer programs was firstly presented. Back in the 1990s, the authors introduced the genetic breeding of nonlinear optimal control strategies for the broom balancing problem in [34]. They considered a function set of basic arithmetic operations, and the system’s model’s states’ variables, such as terminals, and a fitness function based on a linear approximation that replaces the hyperbolic functions in terms of the Taylor series expansion. Operative controllers were generated that outperformed classical controllers at that time for the specified plant.

More recent pioneering investigations, such as those by [35,36], describe the automatic controller generation applied to the vehicle field. Having as plants (i) a combination of an electric motor and electronic drive and (ii) an active suspension system, it is described how it is possible to obtain valid control expressions using evolutionary algorithms. Moreover, out of these examinations, it is noted that, among the results, some were able to manage nonlinearities, despite their linear nature. For instance, in the case of active suspension systems, the results included asymptotically stable candidates, which become unstable if a large bump is encountered and are thus lacking robustness. Following this line of investigation, research in [37,38,39] integrated GP into the field of mobile robotics control. These studies demonstrate the ease with which custom operations can be integrated into the function set used to create the individuals. In these cases, instead of directly obtaining a mathematical equation as the regulation structure, the controller is an algorithm that combines arithmetic, trigonometric, and custom-made functions. Since the algorithm employs user-defined functions, the resulting controller can be understood by the researchers and, if necessary, transformed into a mathematical equation.

Two major trends can be discerned in the field of automatic controller generation using genetic programming. The first one involves the generation of control structures tailored to specific plants or systems. For instance, research developed in [40] aims to simultaneously generate four controllers for a helicopter to perform hovering maneuvers; studies in [41] describe the automated synthesis of optimal controllers using multi-objective genetic programming for a two-mass-spring system; in [42], the objective is about the control of a turbulent jet system; and work in [43] seeks a control structure for a two-dimensional version of the Goddard rocket problem. In all these studies, despite dealing with complex plants or systems, their function sets only included basic arithmetic operators, like exponential and trigonometric operations. From classic control theory, it is hard to imagine a controller minimizing the tracking error as much as possible without the use of integral terms. Nevertheless, GP algorithms demonstrated an ability to find an appropriate relationship among the specified functions and terminals, which was able to fulfill the control objectives.

In line with this first trend, some works can be also identified that combine elementary mathematical operations and custom-made functions to generate control programs for specific systems. In [44], using an acrobot as a plant, the automatic control generation for its minimum time swing up and balance regulation is described. The work introduces a set of logical operations into the function set. Although the final control structure manages to accomplish the regulation objectives, the combination of logical and mathematical operations results in a long expression that can be treated as a control program instead of a control equation. Similarly, research detailed in [45] describes the generation of controllers for high-level applications on a service robot. In this case, the set of functions considered for the evolution process is fully composed of nine custom-made functions, which are unique for this system. The controllers obtained from the work are lists of actions, which can be treated as a program, too. As in many analogous cases, the generated controllers outperformed the baseline alternatives.

The second major trend being highlighted is the research on the generation of control structures for generic systems or groups of plants. In contrast to the previous research line, the function set is restricted to simple mathematical operators. The choice of basic elements for the function set is driven by the main objective of these algorithms, which is to find a suitable mathematical expression that can solve the problem and provide an idea on how to deal with similar scenarios. Research in [46] is a good example for this trend, where authors are looking for the generation of optimal controllers for linear and non-linear plants. In this case, a variant of GP, known as archived-based GP, is used as an evolutionary strategy. It is characterized by the use of additional evolutionary functions, such as piling, sorting, excerpting, production, and archiving. The function set described in this work consists of the four basic arithmetic operations. Corresponding to this trend, the work presented in [47] uses linear GP to generate controllers for nonlinear dynamics with frequency crosstalk. Unlike the previous article, trigonometric operations are also included in the function set. In [48], a technique named multiple basis function genetic programming (MBFGP) is proposed. The structures of program trees in MBFGP are composed of a random number of linear and/or nonlinear basis functions (terms), which are forced to be linear in parameters. The function set used for the evolutionary process is composed of arithmetic operations.

In the previously mentioned research works, controllers for generic structures are generated using GP techniques relying on simple mathematics operations, avoiding the use of differential operations such as integrals and derivatives to obtain operative structures. However, these GP techniques compensate the lack of differential operators by increasing the length of the control expression significantly, attempting to mimic pseudo integral and derivative components. Our starting hypothesis is that differential operators are a pivotal component in an optimized control structure. Their use is expected to greatly enhance the regulation performance while reducing the expression length.

This research approach aligns with the work presented in [49]. There, second- and third-order generic systems are included into the function set of an evolutionary algorithm responsible for generating human-competitive controllers. Controllers designed with this algorithm in the Laplace domain are reported to outperform PID (proportional–derivative–integral) control structures. Another contemporary work worth mentioning is [50], where derivatives and integrals are also included in the function set. Once again, the control structures obtained from this research show a better performance than those designed using traditional methods, such as those guidelines in [51]. In this case, the controller is also developed in the Laplace domain.

Previous research has also used differential operators in genetic programming (GP) in different forms. In [52], the evolutionary process is addressed towards the generation of control structures in the form of block diagrams operating in the Laplace domain. Similarly, research is described in [41] about the integration of differential operators into GP-generated control structures. Although only basic arithmetic operations are used in the function set, the authors included the Laplace operator into the terminal set. This methodology allows the generation of structures in the form of transfer functions.

As has been described, the integration of differential operations into GP-generated control structures in the available literature always involves designing in the Laplace domain. Working in the Laplacian domain ensures the creation of structures that, in one way or another, will have integral and derivative components. Moreover, the evolutionary paradigm allows for the avoidance of human preconceptions regarding control design that are not exploited in current approaches. However, current approaches restrict the range of possible solutions to PID-type controllers. Nevertheless, the most significant limitation of these methods is their ability to produce functional expressions that may seem viable in the Laplace domain but are impractical in the physical world due to their complexity, preventing their transformation into the time domain.

To overcome the previous limitations, this paper proposes a GP framework that operates within the second trend, that is, designing and tuning generalist controllers, and performs the controller generation, design, and tuning in the time domain while simultaneously using differential operators. Moreover, the designed controllers are obtained by minimizing a fitness function that gathers merits such as standard control performance specifications and includes additional requirements, such as robustness against unmodeled dynamics or external perturbations, as well as the implementability of the solutions.

With this approach in mind, our goal is to design a GP-based evolutionary strategy capable of creating control expressions that use differential operations from the function set rather than the terminal set, while at the same time all of the design and tuning process is completed in the time domain. Theoretically, as will be demonstrated, operating in this manner will fully expand the group of possible combinations that the computer can present as control solutions for complex plants, and, importantly, all of these solutions will be fully realizable.

## 4. Automatic Controller Generation, Design, and Tuning

We propose the design of a genetic programming (GP) procedure for the automatic generation, design, and tuning of controllers. The GP-based evolutionary strategy is capable of generating tuned control expressions that include differential operations within their structures, with the overall design process taking part in the time domain. To achieve this challenging result, the function set, F, and the terminal set, T, which constitute the controller structure, must be defined. Moreover, a fitness function, J(·), must be defined, enclosing both the desired behavior for the controlled process and the control action. These elements will serve to obtain an optimal controller using a general GP evolutionary algorithm.

In this research work, a control structure will be considered optimal when meeting the following five requirements:R1.Small-sized expression for the controller;R2.The control action is as smooth as possible;R3.The control action is robust and can withstand variations in the system’s dynamics;R4.The process variable does not surpass a specified overshoot;R5.The steady state error is minimal after a specified time.

Regardless of the GP algorithm employed, the five specified requirements must be contained in a fitness function allowing the GP-based procedure to generate, design, and tune time-domain optimal control strategies in the time domain for general SISO (single-input, single-output) systems.

### 4.1. GP-Based Controller Generator

In order to develop a GP evolutionary strategy satisfying the aforementioned five requirements (R1–R5), special attention is dedicated to the definition of the function set, the terminal set, and the fitness function. The function set should include time-domain differential operators, the terminal set should allow time-domain closed-loop simulations during the optimization phase, and the fitness function bears the ultimate responsibility of identifying controller equations fulfilling all five requirements.

#### 4.1.1. Function Set

According to our approach, the function set, F, is divided into two groups of operators based on the number of inputs they accept. The first group contains bi-variate operators, including the following operations: addition, subtraction, multiplication, and division. The division operator is configured to handle division by 0 by performing a division by 0.001 when such a situation appears during the evaluation of the tree expression. The second group of functions corresponds to uni-variate operators. It includes the operations of exp, log, $\log_{10}$, sin, cos, integral, and derivative. Similar to the division operator, the logarithmic operators are modified functions that return 0 in the case of input of 0. It is worth noting that all the mathematics of these functions is carried out in the time domain, including the differential operators.

#### 4.1.2. Terminal Set

In a similar form, the terminal set, T, must be defined. In this research, the terminal set consists of system signals: states $\left\{x_{0},\ldots ,x_{n}\right\}$, output *y*, control action *u*, reference signal *r*, and error ε. Additionally, the constant *k* is included to allow for the integration of operations with constant parameters and weighted functions. Another issue to be taken into account for the set-up of the GP-based procedure is the determination of initial values for the probabilities associated with the evolutionary process and their bounds (see Table 1): mutation, crossover, and top individuals.

#### 4.1.3. Fitness Function

The fitness scores of the individuals of a given population are calculated based on the minimization of the cost function defined in Equation (Equation 1):(1)J(CA,RI,EL)
which is based on three parameters related to the five requirements indicating the optimality of the solution:The score of the controller action, denoted as CA, related to requirements R2, R4, and R5, which has to deal with the shape of closed-loop dynamics.The robustness index of the controller, named RI, which is linked to requirement R3, which may include unmodeled dynamics or external perturbations.The control expression length, represented as EL, related to requirement R1, which concerns the feasibility of implementation.

To obtain the required data, each controller candidate is simulated based on the state matrices of the plant. These values can either be directly specified in the configuration file or obtained from a plant defined as a transfer function using Equation (Equation 2):(2)$$\begin{array}{ccc} \dot{x}\left(t\right) & = & A(t)x(t)+B(t)u(t)\\ y(t) & = & C(t)x(t)+D(t)u(t) \end{array}$$

Hence, the cost function in Equation (Equation 1) could be expressed in the form of Equation (Equation 3):(3)$$J\left(\cdot \right)=J_{\dot{x}\left(t\right),y\left(t\right)}\left(\cdot \right)$$

The score of the controller action, CA, is the result of the addition of two weighted components, inspired in standard optimal control theory, defined as follows:(4)$$CA=Q\int t\cdot \varepsilon^{2}\left(t\right)dt+R\int u{\left(t\right)}^{2}dt$$

The first component in Equation (Equation 4) describes the reference tracking speed using the ITSE (integral time squared error) criterion [53]. It is defined as the integral of the error multiplied by its time component raised to the power of two. This part of the score is weighted by a constant *Q*. The second component describes the control action speed using the integral of the squared control action. This part is weighted by a constant *R*. The purpose of the constants *Q* and *R* is to assign weights to errors and control actions, typically according to their fabrication and energy costs.

As additional means to calculate the control action score, there are supplementary indicators that modify its final value to allow penalizing deviations from the specified control performance requirements:(5)$$\begin{array}{cccc} if & \left(\left|\varepsilon \right|>0\right)\;\&\;\left(t>stab\_time\right) & then & CA=3\cdot CA\\ if & \left(max\_overshoot<y\right) & then & CA=2.5\cdot CA\\ if & max\left(\left|\frac{d^{2}}{dt^{2}}\left(y\right)\;\right|\right)<signal\_slope & then & CA=2\cdot CA \end{array}$$
where limit values in the formulation (Equation (Equation 5)) are described in Table 2.

The variables of these indicators are related to the occurrence of three events:The tracking error is not minimal after a time value specified by the variable stab_time after a reference change;The system’s output overshoot exceeds the value specified by the variable max_overshoot when tracking a reference change;The maximum value of the second derivative of the system’s output signal does not exceed the limit specified in the variable signal_slope.

The last component is integrated to control the slope of the plant’s output and prevent abrupt signal variations that might affect the useful life of the system’s actuator in a physical implementation. If any of these events occur, the value of the control action score will be multiplied by 3 in the case of tracking error, 2.5 in the case of a high overshoot, and 2 in the case of rough system output.

When using our genetic programming procedure, both structures and their parameters are optimized according to the fitness evaluation method. In this investigation, the equations generated during the evolutionary process determine the structures, while the constants that modify the weight of the mathematical operations in the equations are represented by the parameters. Within the proposed genetic strategy, the structures are optimized using the natural evolution process of the algorithm. The parameters are fine-tuned (optimized) using the standard Nelder–Mead optimization method [54]. As expressed in the available literature, both components of a valid individual could be simultaneously processed in the evolutionary process; however, this approach would lead to a huge search space for the GP system. Additionally, in the context of automatic controller generation, joint optimization for design and tuning could potentially lead to controllers with a satisfactory performance and stability due to the tuned values of their constant parameters. However, it may result in a poorly designed structure that lacks robustness. This issue emerges because the generated controller is a unique solution for the dynamic equation used in the simulation process. As a result, minimal changes in one of the equation’s coefficients might mean that the controller will no longer be a functional structure.

In the available literature, it is a common trend to simulate the combination of a controller plus the plant operation to obtain analytical data for fitness scoring. However, this is typically performed assuming that the plant’s dynamics are ideal and remain constant all the time. Real-world plants are not ideal and change over time due to aging components and continued usage. Mathematically, these variations do not add or remove factors from the plant’s dynamic equation, but they do alter the values of its coefficients. To simulate this dynamic variation in the plant, a parameter named *_sstep* is used to generate a second plant equation with a slight variation. This modified equation is used in a second simulation process of the fitness calculation. Consequently, two control action scores are obtained at the end of the simulation stage. These values are used in the calculation of the robustness index RI, as described in Equation (Equation 6):(6)$$RI=\frac{\left|{CA}^{\prime }-CA\right|}{\_sstep}$$

Finally, the last fitness parameter of the cost function is the expression length, EL. It is calculated according to the number of levels that a genetic tree has:(7)EL=|GP_tree|
where |·| represents the length of the genetic programming tree denoted as *GP_tree*.

### 4.2. GP Evolutionary Algorithm

A regular and straightforward evolutionary algorithm is considered in our genetic programming approach. The initial stage of the GP procedure involves the generation of the first population of individuals, referred to as generation 0, denoted as $G_{0}$. During this step, elements from the function set, as well as elements from the terminal set, are randomly selected and assembled into GP tree structures.

Once all the individuals (controllers) from a population have been assigned their corresponding fitness scores, the next step in the evolutionary process is to sort them in descending order of fitness according to their cost function values. Then, the generation of a new offspring of controllers is processed using the described genetic operations: elitism, mutation and crossover. The elitism operation is controlled by the parameter *hof_num*, which specifies from fittest to least fit how many candidate controllers are directly copied in the next population of available slots. In the case of the mutation, and crossover, the vectors *mut_lims* and *cross_lims* contain the upper and lower bounds for the probabilities associated with the evolutionary algorithm components.

Balancing exploration (mutation) and exploitation (crossover) in an evolutionary process is often challenging. Most of the time, a set of probabilities that leads to good results in one scenario does not generate the same favorable results in different ones. The literature suggests that during the initial stages of an evolutionary process, it is usually more effective to prioritize exploration over the exploitation of existing individuals. As the process goes on, a group of candidates with good performances appears, which makes it better to exploit local individuals rather than to keep looking for new alternatives. Initially, the mutation probability exceeds the crossover probability. However, as generations progress, the mutation probability decreases, while the crossover probability increases, following linear relationships. These variations in rates are calculated using the expression in Equation (Equation 8):(8)$$VarRate=\frac{maxLimit-minLimit}{GenerationsNumber}$$

The evolutionary algorithm uses genetic operations to fill about 90% of the defined number of individuals for every population. The remaining 10% of individuals are generated using the controller generator defined for the initial population creation. This approach facilitates the continuous integration of new components into the evolutionary process, which might lead to an overall generation improvement.

## 5. Implementation

To implement the proposed GP-generator procedure, a Python-based GP program has been developed. This code can generate time-domain optimal control strategies for general single-input, single-output (SISO) systems. The programming is coded in Python 3 and uses the open-source package DEAP, which stands for Distributed Evolutionary Algorithms in Python [55]. DEAP is an evolutionary framework that allows for the fast development, implementation, and testing of code related to evolutionary techniques. It includes a range of predefined structures, objects, and functions that encapsulate the basic components and operations inherent to an evolutionary process. Additionally, it offers easy integration of custom-made components, allowing compatibility with any kind of development.

The final source code for the proposal is composed of three main files:The configuration file, which contains the values and definitions of variables used throughout the system’s execution.The custom-made Reverse Polish Notation (RPN) calculator library.The main file, which encodes the evolutionary strategy and report generator.

The system’s operation is visually represented in Figure 2.

### 5.1. The Configuration File

The configuration file specifies the state-space matrices of the system to be controlled, either by directly specifying their values or by calculating them and specifying the system’s transfer function H(s) instead of the matrices. The program is coded to automatically detect how the plant has been specified and determine whether it is necessary to calculate its state-space matrices. In addition to this information, the configuration file also contains the following: (i) the system’s simulation variables, (ii) limit values and execution constants for the fitness calculation, and (iii) variables and probabilities for the evolution process. For those variables that have not been previously defined in earlier sections, a list is provided in Table 3.

### 5.2. The RPN Calculator

To date, both customized and generalist procedures for applying GP in control prevent the generation of controllers endowed with differential operators designed in the time domain due to complexity issues.

To overcome the coding complexity issue, our approach develops and implements a custom-built Reverse Polish Notation calculator implemented in Python 3. RPN is a method for conveying mathematical expressions without the use of separators such as brackets and parentheses. In RPN, operators follow their operands, hence removing the need for brackets to define evaluation priority. The operation is read from left to right, but execution is performed every time an operator is reached. The RPN procedure always uses the last one or two numbers as operands. This notation is suited for computers and calculators since there are fewer characters to track and fewer operations to execute. The main objective of this custom RPN calculator library is to perform mathematical computations required for system simulations included in the fitness evaluation. RPN notation was selected over Infix notation (commonly used in arithmetical and logical expressions) for simulation calculations because, when combined with a memory structure, RPN’s stack-based evaluation provides efficient and organized access to the memory positions of the differential operators memory structure. Hence, the code complexity for these operators is reduced to the minimum.

### 5.3. The Main File

When the required set-up parameters are loaded, the main program proceeds with two processes. First, it generates a Python pickle structure, and, simultaneously, it initializes an empty PDF document for historical data storage and report generation. Next, the initial population of individuals is generated. The function in charge of this procedure randomly selects mathematical operations and terminals from predefined sets and combines them into GP tree structures.

To obtain values for the variables involved in the different calculus of the fitness function, each controller candidate is simulated twice using the RPN calculator library for mathematical operations. The system’s simulation for the feedback control of the plant is performed using the state-space equations described in Equation (Equation 2). The combined use of the library and the equations allows for an iterative, step-by-step simulation of the system. As a result, the verification of overshoot and stabilization parameters can be completed on the fly during system operation. It also brings the advantage that if a value exceeds a predefined maximum limit during the simulation, the process is halted, and all scores are assigned an infinite value for the fitness parameters of a control candidate, leading to its immediate discard from the evolution process. In the case that either the simulation process runs to completion or the maximum number of generations, denoted as *gen_number*, is reached, historical data on error, control action, and the system’s output are processed to calculate the value of the controller action score.

## 6. Tests and Results

Most of the plants commonly encountered in practical control applications can be adequately approximated by either a first-order or a second-order transfer function, simplifying the control design process. Therefore, it will be assumed in this research that restricting the considerations to these types of plants provides a valid testing scenario.

In the classic control approach, a notable case is reference tracking. When the control objective consists of making the system follow a constant reference signal, a common answer is to use a proportional controller based on the reference signal and the system’s gain. As mentioned, if the reference remains unchanged over time, the tracking performance of this alternative is flawless. However, if this strategy is faced with varying references, the tracking performance deteriorates. To eliminate the possibility that the program’s outcome generates controllers relying solely on this structure, all validation design tests use a reference signal with three variations over time. The values were selected to have a small, a medium, and a large variation, thereby preventing the generation of controllers that can only handle a specific range of reference signal variations.

The test suite is designed to demonstrate three fundamental aspects of the algorithm’s features:1.It possesses the ability of generating functional control algorithms using the mathematical operations defined in the function set.2.It can handle complex control systems.3.It can generate controllers for both first-order and second-order plant models.

Referring to the first aspect, a practical approach involves modifying the code and seeing if the program is capable of generating structures that replicate or are similar to the classical PID equation structure. For this work, the previously mentioned modifications are to limit the elements of the function set to the four basic math operations (+,−,∗,/) and the differential functions ($\int ,\frac{d}{dt}$). In the case of a successful outcome, this scenario would validate the effectiveness of the evolutionary algorithm, confirming that the program is able to combine the elements of the function set to meet the objectives defined by the fitness function.

According to control theory, a system is considered unstable when any of its natural poles has a positive sign. Additionally, a plant may pose control challenges when it exhibits a very high natural gain, with minimal input values causing significant variation in the plant’s outputs. These characteristics can be used as instruments to assess how the algorithm performs when dealing with complex systems. Therefore, the test suite requires the program to generate control equations for first-order plants with positive poles. Furthermore, the plants in the test scenarios will exhibit a range of gain values, including small, medium, and large gains. Finally, once the two initial statements are proven true, a final series of tests will be carried out to assess the algorithm’s performance when facing second-order systems with negative poles.

As a result of the described planning, the following tests are proposed to validate the GP-based controller generator:First-order plant, negative pole, low gain, and using the reduced funtions set (RFS);First-order plant, negative pole, medium gain, and using the RFS;First-order plant, negative pole, high gain, and using the RFS;First-order plant, negative pole, and low gain;First-order plant, negative pole, and medium gain;First-order plant, negative pole, and high gain;First-order plant, positive pole, low gain, and using the RFS;First-order plant, positive pole, medium gain, and using the RFS;First-order plant, positive pole, high gain, and using the RFS;First-order plant, positive pole, and low gain;First-order plant, positive pole, and medium gain;First-order plant, positive pole, and high gain;Second-order plant and low gain;Second-order plant and medium gain;Second-order plant and high gain.

To ensure a consistent performance evaluation, all evolutionary processes use the same configuration parameters, as listed in detail in Table 4.

Finally, the test set is categorized into groups based on the system’s order and the nature of its poles, resulting in the following groups:Group 1: Tests 1–6. First-order stable systems.Group 2: Tests 7–12. First-order unstable systems.Group 3: Tests 13–15. Second-order systems.

### 6.1. First-Order Stable Systems

The plant functions used as targets for the genetic evolution are depicted in Equation (Equation 9). As can be seen, each test increases the plant’s gain by a factor of 10.
(9)$$H\left(s\right)=\frac{K}{0.5s+1}\;,\;K=\left\{0.4,4,40\right\}$$

After completing all the tests in the group, the data corresponding to the best individuals for each case are shown in Table 5. It consists of 4 columns, and from left to right, it describes (i) the generation number when the expression was found, (ii) the raw expression of the controller, (iii) the equation expressed using mathematical symbols, and (iv) the final *J* cost value of the control candidate.

The results for Tests 1–3 in Table 5 show that all the best controllers are expressed as weighted integrals of the error signal. It is worth mentioning that the evolutionary algorithm was efficient in generating these solutions. Based on the generation numbers where the controllers were found, it can be seen that it took only half of the available generations to find functional solutions. In fact, the integral structure for each of these three cases was identified as the best structure even in earlier generations: generation 3 in Test 1, generation 5 in Test 2, and generation 8 in Test 3. Following these points in the evolution process, the algorithm was mainly trying to optimize the equations’ constants as much as possible.

When checking the graphical representations of the equations for Tests 1–3, shown in Figure 3a–c, it can be seen how the controllers for Test 1 and Test 2 exhibit fast stabilization, a minimal overshoot peak, and, after the initial peak, a flawless tracking performance for all reference variations. However, in the case of Test 3 the behavior depicted in the image displays three oscillations after reaching a stable tracking state, which is the result of controlling a plant with an excessively high gain.

Regarding the results of Tests 4 and 5, Table 5 illustrates that these equations no longer consist of pure integral operations. In the case of Test 4, the equation now shapes a proportional–integral (PI) structure. Its proportional part is based on the weighted error and reference. As a result, it can be observed in its graphical behavior shown in Figure 3d that this structure is more aggressive. The stabilization time is the fastest of this group, the overshoot is minimal, and the reference tracking is again flawless. For Test 5, the control equation includes a derivative term. However, it is the derivative of the time variable, which equals 1. Therefore, after simplifying the equation, we have a weighted integral structure again. The corresponding graph in Figure 3e reveals that this controller is the most accurate one. Its behavior has no overshoot, a fast stabilization time, and excellent reference tracking.

The most challenging case within this group is Test 6, where the objective is to control a plant with an exaggerated gain. As described in Table 5, the best equation found by the program contains an exponential element and was discovered at generation 32. From its graphical behavior, depicted in Figure 3f, it can be observed that for a small reference variation the equation describes 10 oscillations before stabilizing. However, for medium and large signal variations these oscillations were drastically reduced. Given the program’s ability to design controllers for plants with a large gain and RFS and the previous cases that used the full function set, it can be inferred that in this particular case 50 generations were insufficient evolution generations for the algorithm to discover and optimize the best individual.

To better understand the evolutionary path of Test 6, details are depicted in Table 6. As illustrated, throughout this process, the algorithm explored various mathematical operations, including exponential and trigonometric functions. Referring to Figure 4, it can be observed that after struggling to fully track the reference until generation 20, at generation 21 it successfully replicated the reference’s pattern. Subsequently, the algorithm attempted to minimize the offset error by experimenting with oscillatory alternatives, ultimately reaching generation 32 and achieving precise signal tracking after a certain period.

A summary of the best controllers in this group is described in Table 7. The columns, from left to right, contain (i) the plant’s equation, (ii) the controller generated using the full function set, and (iii) the controller generated using the reduced function set (RFS).

A summary of the best controllers in this group is provided in Table 7. The columns, from left to right, include (i) the plant’s equation, (ii) the controller generated using the full function set, and (iii) the controller generated using the reduced function set (RFS).

### 6.2. First-Order Unstable Systems

The plant functions used as targets for the genetic evolution are depicted in Equation (Equation 10). As can be seen, each test increases the plant’s gain by a factor of 10.
(10)$$H\left(s\right)=\frac{K}{0.5s-3}\;,\;K=\left\{0.4,4,40\right\}$$

Similar to the previous group, the data corresponding to the best individuals are depicted in Table 8. This table follows the same structure as the previous group.

When examining the results corresponding to Tests 7–11 in Table 8, it can be observed that all the individuals include an integral term. In the case of Test 7, the equation exhibits a PI structure, where the proportional term is based on the error, reference, and current state. As shown in Figure 5a, this combination results in fast tracking with a small overshoot peak relative to the reference variation. For Test 8, the table also describes a PI structure. However, this time, the proportional element is only based on the current state. Figure 5b illustrates that this equation generates the best tracking behavior among Group 2. It demonstrates a fast reaction time, minimal overshoot peak, and perfect reference tracking. Finally, for Test 9 the best controller takes the form of a weighted integral of the error. As depicted in Figure 5c, the program was able to find a constant that enables fast tracking with overshoot only for significant reference variations. Its is worth noting that, once again, when using the reduced function set (RFS), the algorithm identified the best individual’s structure in early generations, leaving the rest of the evolution process to optimize the constants.

When using all the operations from the function set, Test 10 shapes a PI structure. The most interesting part of this result is that, in this case, the algorithm includes the previous output value of the plant in the control equation. In Figure 5d it is shown that this structure presents fast action and overshoot peaks in response to large reference variations. The individual found for Test 11 also describes a PI structure, with a proportional part relying on the current state. By observing Figure 5e, it can be seen that the behavior of this equation closely resembles the previous test. However, once again, when dealing with the most challenging problem in this group, the program managed to discover an equation that replicates the reference’s base outline but fails to eliminate the offset error, as seen in Figure 5f. In this case, the optimal controller includes derivative and exponential operations. To gain a deeper understanding of the evolutionary path of this test, refer to Table 9.

A deeper analysis of Test 12 reveals that throughout the evolutionary process the algorithm is not able to incorporate an integral operation. As seen in Figure 6, during its initial stages the algorithm experimented with equations involving trigonometric operations, with limited success. Once the evolution path got rid of them, it could be seen that the new equations have the ability to replicate the reference’s outline. The last significant leap in this evolutionary path took place at generation 47, yet this structure still had the offset issue and a substantial overshoot prior to stabilization. Similar to the previous group, it can be inferred that once again the range of 50 generations was insufficient for the algorithm to discover a functional equation, given the complexity of the plant.

The final equations of the second group of tests are summarized in Table 10 along with their respective plant equations.

From the previous groups of tests, the results confirm that the proposed algorithm is capable of generating control equations for complex systems. Depending on the systems’ complexity, an increased number of generations is needed to find a suitable solution. Furthermore, it is also proved that the program can address control problems related to first-order plants.

### 6.3. Second-Order Stable Systems

The plant functions used as targets for the genetic evolution are depicted in Equation (Equation 11). As observed, each subsequent test increases the plant’s gain 10 times.
(11)$$H\left(s\right)=\frac{K}{s^{2}+1.5s+0.5}\;,\;K=\left\{0.35,3.5,35\right\}$$

With the results obtained in the previous subsections, the only remaining statement that needs to be proven true is the one corresponding to “the algorithm can also handle second order equations”. Therefore, in this case there is no need to increase the complexity of the second-order system to exaggerated levels. This group contains only plants with negative poles. However, systems with low, medium, and high gains are used to observe how the algorithm reacts. The results of the tests are described in Table 11.

When checking the final equation of Test 13, it can be seen that the equation corresponds to a PI structure. It is worth noting that this equation was found in generation 30. Figure 7a depicts how the tracking action of the controller draws an overshoot peak during its operation. In the case of Test 14, the equation has the best performance of the group. It presents a PI structure where the proportional part is based on the current value of state 1. Figure 7b shows how the tracking action is fast but not aggressive, has no overshoot peak, and has minimal steady-state error. Finally, the equation of Test 15 describes a PI form. The proportional part is based on the error and both of the system’s states, while its integral component is the result of a division of time by the weighted integral of the error with the addition of the system’s state 1. In Figure 7c, it can be seen that this equation has the most aggressive behavior of the group. However, its overshoot peak is minimal.

After analyzing the results of this group, it is proven true that the proposed algorithm is capable of generating controllers for both first-order and second-order plants. This means that a GP-based controller designer has been successfully generated, and it has the ability to generate operational controllers even for complex plants.

## 7. Conclusions

This research work has successfully attained its primary objective of developing a genetic programming-based controller generator. The proposed system represents a noteworthy achievement, showcasing its ability to design effective control structures whose structures have tuned and optimized individual parameters, allowing for the regulation of first-order and second-order single-input single-output (SISO) plants. The comprehensive testing conducted throughout this study demonstrates that the genetic programming (GP)-based approach holds promise in addressing complex control problems that often surpass the capabilities of traditional, human-driven methods.

One significant advantage of the proposed generator lies in its capacity to create control structures in the time domain. This design choice brings several benefits, including the direct incorporation of differential operators such as integrals and derivatives within the equation structure. Additionally, it eliminates the generation of control candidates that cannot be practically implemented in control devices due to challenges associated with inverse Laplace operations. The result is a streamlined process where every proposed controller can be seamlessly implemented in control devices without encountering coding complications.

The experiments conducted yielded compelling results, showcasing the algorithm’s capability to produce optimal control structures. These structures not only meet key criteria, such as fast error reduction, rapid control action, minimal overshoot, null error tracking after a specified time, and a compact equation length, but also excel in fulfilling them. It is noteworthy that these successful outcomes extend across both first- and second-order plants, which are prevalent structures within industrial settings.

While specific examples, such as second-order oscillatory systems, integrating systems, and delayed systems, were not explicitly covered as test subjects, the results from our comprehensive testing reveal the controller designer’s capability to handle a broader spectrum of complex and diverse plant types. The key point is that, given a sufficient number of generations to evolve, the controller designer exhibits adaptability and potential for dealing with various system complexities.

The conducted series of tests demonstrate that the controller generator possesses the inherent capability to fulfill its evolution objectives. This effectiveness is constrained only by the configuration of the genetic evolution process. A notable strength of employing GP for obtaining control equations is the inherent nature of the evolution methodology. Over successive generations, the GP approach naturally sifts through various function and variable combinations, allowing the most effective ones to prevail. This iterative process ensures that if the control problem’s solution involves incorporating differential operations in the equation structure, the algorithm will eventually discover combinations that successfully address the problem.

In essence, the use of GP provides a dynamic and self-adapting framework. The evolutionary process enables the identification and selection of optimal functions and variables, making it a powerful tool for solving intricate control problems. When confronted with more complex plants, the challenge lies in determining the appropriate number of generations. This decision is crucial as it provides the designer with the necessary range to thoroughly explore and exploit potential solutions. In conclusion, the complexity in dealing with more intricate plants is tied to the right selection of the generation number, ensuring a robust exploration of the solution space. This approach allows the GP-based controller designer to navigate and conquer the challenges posed by diverse and complex plant dynamics.

It is important to recognize that the observed success in these initial behaviors serves as a foundation for future developments, particularly with higher-order plants. The scalability and adaptability of the proposed genetic programming-based controller generator open avenues for further exploration and application in more intricate control scenarios, such as nonlinear and multi-input multi-output (MIMO) systems and even distributed control. The potential for addressing higher-order plant complexities positions this research as a stepping stone toward advancements that could significantly impact the field of control systems.

## Figures and Tables

**Figure 1 sensors-23-09731-f001:**
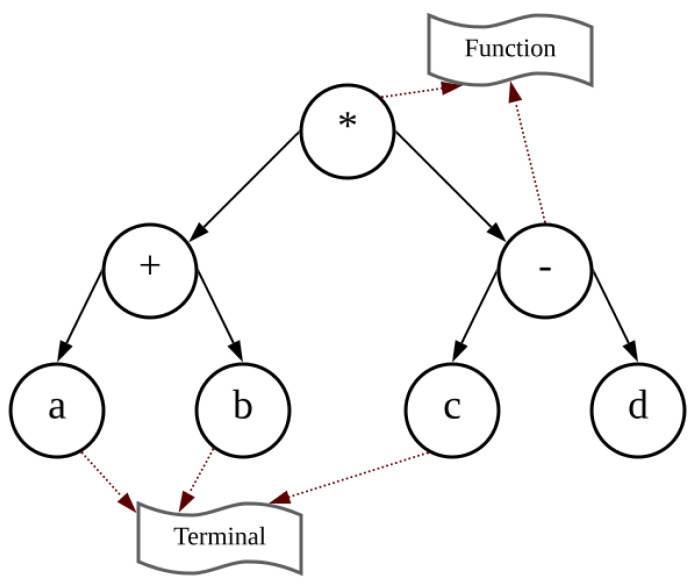
Genetic tree structure in genetic programming. This is the tree representation of the equation (a+b)∗(c−d).

**Figure 2 sensors-23-09731-f002:**
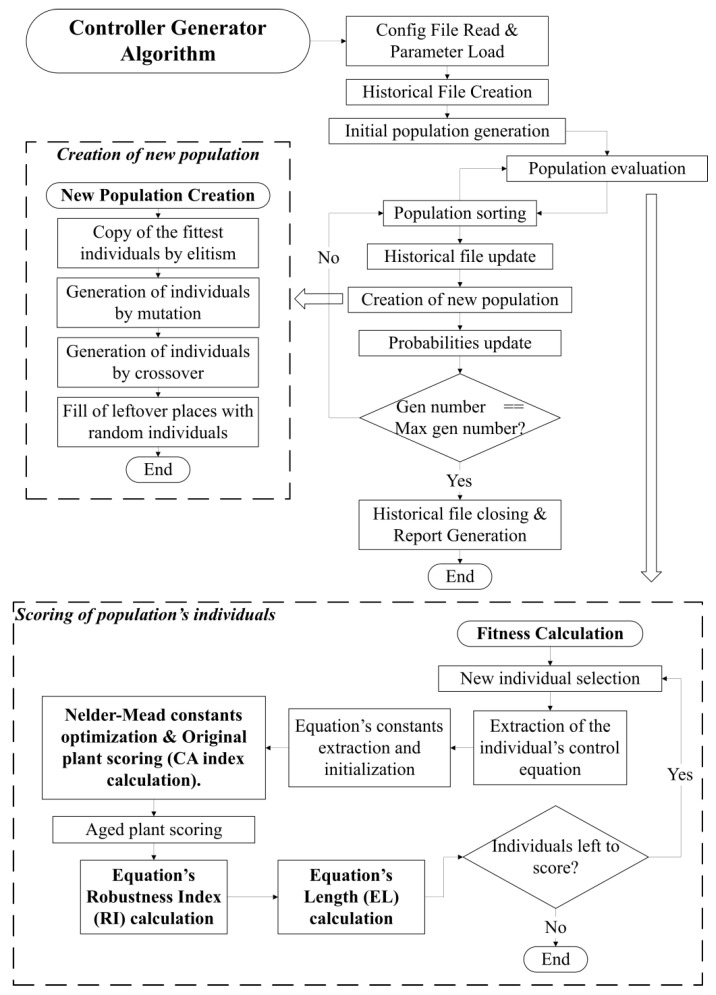
Flow diagram of the final GP-based controller generator.

**Figure 3 sensors-23-09731-f003:**
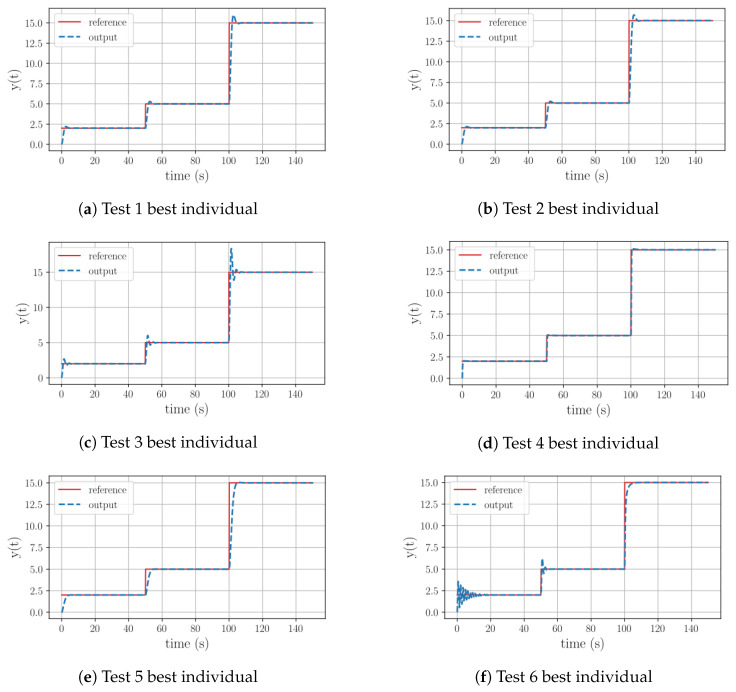
Graphical behavior of the best individuals resulting from the Group 1 test set.

**Figure 4 sensors-23-09731-f004:**
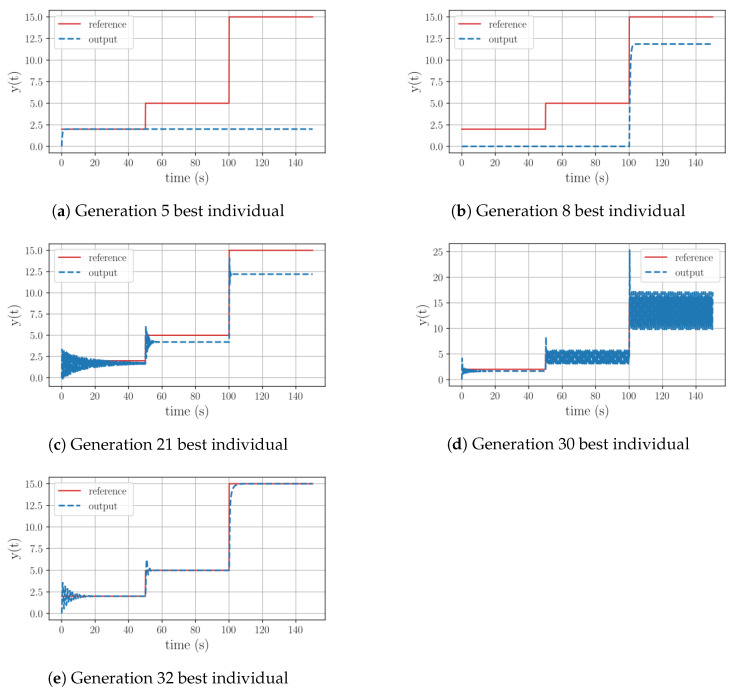
Test 3: Controller’s evolution process for a first-order plant with a negative pole with a high gain, using all the available functions to generate equations.

**Figure 5 sensors-23-09731-f005:**
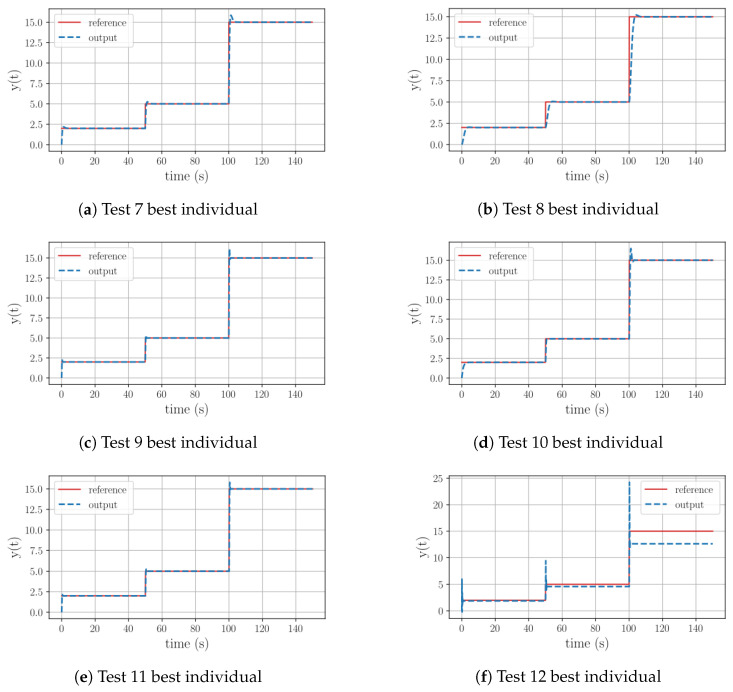
Graphical behavior of the best individuals resulting from the Group 2 test set.

**Figure 6 sensors-23-09731-f006:**
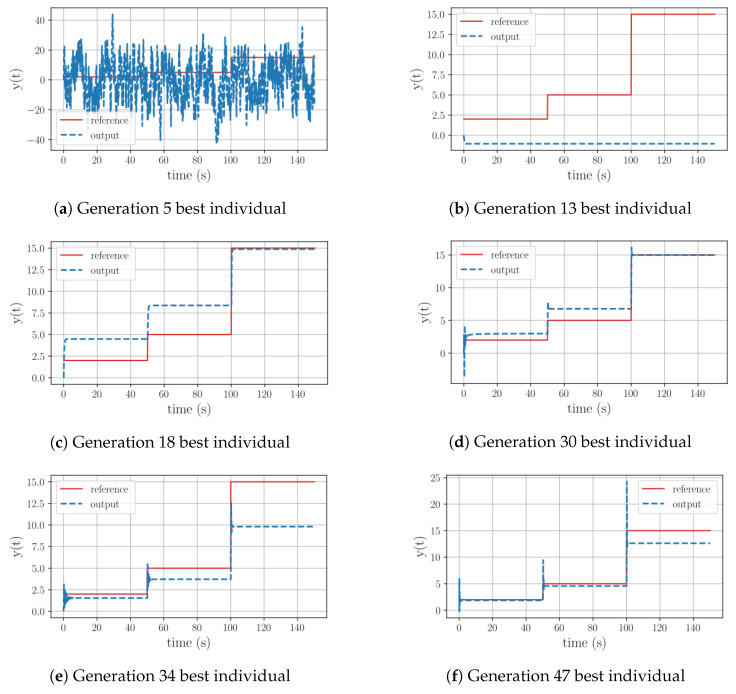
Test 9. Controller’s evolution process for a first-order plant with a positive pole with a high gain, using all the available functions to generate equations.

**Figure 7 sensors-23-09731-f007:**
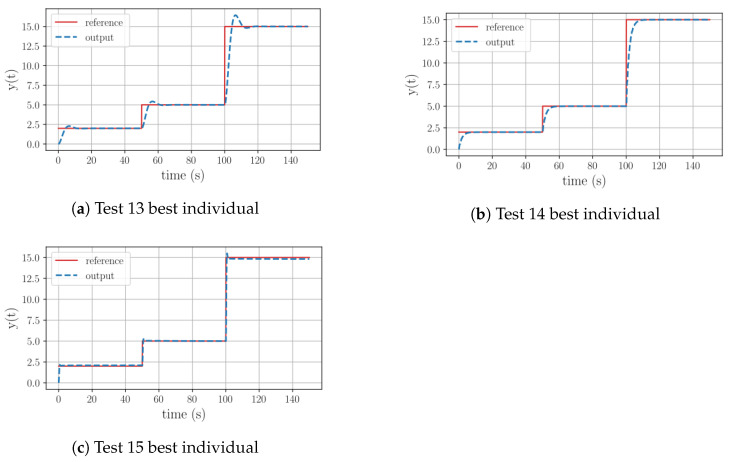
Graphical behavior of the best individuals resulting from the Group 3 test set.

**Table 1 sensors-23-09731-t001:** Set-up of experimental conditions for the GP-based procedure.

Variable Name	Description
*mut_lims*	initial, min, and max values for mutation probability
*cross_lims*	initial, min, and max values for crossover probability
*hof_lims*	number of top individuals copied for the next generation

**Table 2 sensors-23-09731-t002:** Set-up of experimental conditions for the controller action.

Variable Name	Description
*stab_time*	max stabilization time in output for reference variation
*max_overshoot*	max overshoot in the output signal tracking (%)
*signal_slope*	max slope variation of the system’s output

**Table 3 sensors-23-09731-t003:** Set-up for system’s simulation, evolutionary procedure, and optimization.

Variable Name	Description
*dt*	step size for system’s simulation
*st*	stop time for system’s simulation
*refs*	system simulation’s reference values
*maxtol*	max tolerance in optimization process
*maxiter*	max number of iterations in optimization process
*ind_number*	number of individuals
*gen_number*	max number of generations

**Table 4 sensors-23-09731-t004:** Configuration parameters used in the test suite.

Variable Name	Value
*_sstep*	0.3
*dt*	0.1
*st*	150
*refs*	[2, 5, 15]
*max_overshoot*	0.1
*stab_time*	10
*signal_slope*	10
*ind_number*	120
*gen_number*	50
*hof_lims*	1
*cross_lims*	(0.31, 0.69)
*mut_lims*	(0.35, 0.64)

**Table 5 sensors-23-09731-t005:** Group 1 list of best individuals of each test.

Gen	Raw Expression	Equation	*J*
**Test 1**			
13	kinte(e0)	2.78∫ε(t)dt	$2.03\times 10^{6}$
**Test 2**			
22	kinte(e0)	0.24∫ε(t)dt	$2.20\times 10^{6}$
**Test 3**			
27	kinte(e0)	0.06∫ε(t)dt	$1.12\times 10^{6}$
**Test 4**			
17	add(kinte(e0), add(e0, add(add(r0, add(e0, e0)), r0)))	3ε(t)+2ref+1.92∫ε(t)dt	$6.44\times 10^{5}$
**Test 5**			
49	div(kinte(e0), add(x0, add(y0, kderi(t))))	$\frac{0.15\int \varepsilon (t)dt}{\frac{d}{dt}\left(t\right)}$	$2.19\times 10^{6}$
**Test 6**			
32	div(e0, kexp(inv(kexp(e0))))	$\frac{0.34\int \varepsilon (t)dt}{e^{x(t)}}$	$5.50\times 10^{7}$

**Table 6 sensors-23-09731-t006:** Tests 6 evolution process. First-order plant, negative pole, high gain, and RFS.

Gen	Raw Expression	Equation	*J*
5	div(inv(kexp(div(r0, r0))), kexp(x0))	$\frac{0.18}{e^{1+x(t)}}$	$2.03\times 10^{8}$
8	inv(add(kexp(kcos(ksin(r0))), u0))	$\frac{1}{9.18e^{6.85\cos\left(7\sin\left(ref\right)\right)}+u_{k-1}}$	$4.66\times 10^{7}$
21	div(e0, kexp(u0))	$\frac{\varepsilon (t)}{6.79e^{u_{k-1}}}$	$3.20\times 10^{7}$
30	div(kinte(e0), kexp(x0))	$\frac{\varepsilon (t)}{6.01e^{\frac{0.45}{e^{\varepsilon (t)}}}}$	$3.47\times 10^{6}$
32	div(e0, kexp(inv(kexp(e0))))	$\frac{0.34\int \varepsilon (t)dt}{e^{x(t)}}$	$5.50\times 10^{5}$

**Table 7 sensors-23-09731-t007:** Summing-up of the best controller for a first-order plant with a negative pole using the total (middle column) or the reduced (right column) function set.

Plant	Controller	Controller
$H\left(s\right)=\frac{0.4}{0.5s+1}$	ref+3ε(t)+1.92∫ε(t)dt	2.78∫ε(t)dt
$H\left(s\right)=\frac{4}{0.5s+1}$	0.15∫ε(t)dt	0.24∫ε(t)dt
$H\left(s\right)=\frac{40}{0.5s+1}$	$0.34e^{-x(t)}\int \varepsilon (t)dt$	0.06∫ε(t)dt

**Table 8 sensors-23-09731-t008:** Group 2 list of best individuals of each test.

Gen	Raw Expression	Equation	*J*
**Test 7**			
13	sub(div(sub(e0, kderi(x0)), kexp(x0)), x0)	$\frac{ref}{x(t)}+\varepsilon \left(t\right)+ref-x\left(t\right)+2.22\int \varepsilon (t)dt$	$3.38\times 10^{7}$
**Test 8**			
18	sub(kinte(e0), x0)	0.11∫ε(t)dt−x(t)	$2.70\times 10^{6}$
**Test 9**			
30	sub(kinte(e0), add(x0, x0))	0.47∫ε(t)dt	$2.92\times 10^{5}$
**Test 10**			
50	sub(sub(kinte(e0), x0), add(y0, div(x0, r0)))	$7.35\int \varepsilon (t)dt-y_{k-1}-x\left(t\right)$ $+\frac{x(t)}{ref}$	$6.20\times 10^{5}$
**Test 11**			
38	sub(sub(kinte(e0), x0), x0)	4.35∫ε(t)dt−2x(t)	$1.98\times 10^{5}$
**Test 12**			
47	sub(div(sub(e0, kderi(x0)), kexp(x0)), x0)	$\frac{\varepsilon \left(t\right)-0.91\frac{d}{dt}x\left(t\right)}{3.67\cdot e^{x(t)}}-x\left(t\right)$	$3.38\times 10^{6}$

**Table 9 sensors-23-09731-t009:** Test 12 evolution process. First-order plant, positive pole, high gain, and full function set.

Gen	Raw Expression	Equation	*J*
5	sub(kcos(y0), x0)	$1.02\cos\left(y_{k-1}\right)-x\left(t\right)$	$4.26\times 10^{9}$
13	sub(kcos(kcos(x0)), x0)	$1.13\cos\left(1.59\cos\left(x(t)\right)\right)-x\left(t\right)$	$3.47\times 10^{8}$
18	sub(div(sub(r0, x0), kexp(x0)), x0)	$\frac{ref-x(t)}{14.54e^{x(t)}}-x\left(t\right)$	$9.01\times 10^{7}$
30	sub(div(sub(r0, sub(inv(t), e0)), kexp(x0)), x0)	$\frac{ref-1/t-\varepsilon (t)}{15.72e^{x(t)}}-x\left(t\right)$	$2.29\times 10^{7}$
34	sub(div(sub(e0, x0), kexp(x0)), x0)	$\frac{\varepsilon (t)-x(t)}{11.06e^{x(t)}}-x\left(t\right)$	$5.01\times 10^{6}$
47	sub(div(sub(e0, kderi(x0)), kexp(x0)), x0)	$\frac{\varepsilon \left(t\right)-0.91\frac{d}{dt}x\left(t\right)}{3.67\cdot e^{x(t)}}-x\left(t\right)$	$3.38\times 10^{6}$

**Table 10 sensors-23-09731-t010:** Summing-up of the best controller for a first-order plant with a positive pole using the total (middle column) or the reduced (right column) function set.

Plant	Controller	Controller
$H\left(s\right)=\frac{0.4}{0.5s-3}$	long	long
$H\left(s\right)=\frac{4}{0.5s-3}$	4.35∫ε(t)dt−2x(t)	0.11∫ε(t)dt−x(t)
$H\left(s\right)=\frac{40}{0.5s-3}$	long	0.47∫ε(t)dt

**Table 11 sensors-23-09731-t011:** Group 3 list of best individuals of each test.

Gen	Raw Expression	Equation	*J*
**Test 13**			
30	add(kinte(e0), e0)	ε(t)+0.75∫ε(t)dt	$3.09\times 10^{6}$
**Test 14**			
37	sub(kinte(e0), x0)	$16.58\int \varepsilon (t)dt-x_{1}\left(t\right)$	$2.57\times 10^{6}$
**Test 15**			
45	sub(add(add(e0, x1), div(t, add(kinte(x1), x0))), x0)	$\varepsilon \left(t\right)+x_{2}\left(t\right)-x_{1}\left(t\right)+\frac{t}{1.25\int \varepsilon (t)dt+x_{1}\left(t\right)}$	$8.28\times 10^{5}$

## Data Availability

Data are contained within the article.

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
