# Peer review of "Revisiting Classical Controller Design and Tuning with Genetic Programming"

_sensors, 2023, doi:10.3390/s23249731_

Round 1

Reviewer 1 Report

Comments and Suggestions for Authors

Brief summary:

A genetic programming (GP)-based evolutionary strategy for automatic generation, design, and tuning of controllers is proposed. The proposed strategy allows generating of tuned control expressions that include differential operations within their structures. The overall design process takes part in the time domain. The performance and validity of resulting controllers are evaluated by verifying whether the generator can replicate the structure and performance of those produced by traditional controller design methods or,  in some cases, ouperform them.

The proposed approach makes a significant contribution to the field of AI in control engineering

References:

The article seems to be pioneering in the field of AI in control engineering. In its introductory sections, a broad portfolio of  references are surveyed gathering respective knowledge since the 1980’s up to now.

There are 53 references in total which is adequate.

Scientific content:

The submitted article has ambition to become a seminal paper. The presented results are innovative and promising.

Specific comments, questions:

-          The selected process models do not cover all basic types of models (second order oscillatory system, integrating systems and delayed systems are not covered). Do you expect any specific problems when using the proposed GP-based design strategy?

-          Eq.(4) -  there is a mistake (using a squared time does not have any sense):

-          What is the meaning of x(t) in the expressions of control action?

-          How do you decide whether the gain is low, medium or high?

  I highly recommend to accept the article after minor corrections in its present form.

Comments on the Quality of English Language

The English is almost perfect, there are only a few typos or small grammatical errors in the article (at lines 37, 225, 384, 657, 674).

Two paragraphs are mistakenly replicated (lines 101 – 105 replicated at lines 106-111; lines 580 -586 replicated at lines 587-593).

Author Response

We add our answers in a dedicated file

Reviewer 2 Report

Comments and Suggestions for Authors

This paper presents an innovative genetic programming (GP) approach for the automated design and fine-tuning of process controllers. What sets this method apart from other controller design techniques is its capacity to encompass the entire design process within the time domain, which includes handling differential operations such as derivatives and integrals without the necessity of an intermediate inverse Laplace transform. As a result, not only is the design process streamlined, but the controllers produced are also readily applicable in real-world physical systems. Furthermore, it's important to note that the GP function set extends beyond basic arithmetic operators, incorporating a wide range of mathematical operations, including trigonometric, exponential, and logarithmic functions. This diverse set of functions enhances the flexibility and adaptability of the GP-based approach. To assess the performance and validity of the controllers generated through this GP-based approach, an evaluation is conducted to ascertain whether the generator can replicate the structure and effectiveness of controllers created by traditional design methods. In some instances, the GP-generated controllers exhibit superior results. In summation, the GP-based approach represents a promising and automated solution for the controller design process, effectively addressing control-related challenges across various engineering applications. 

In this context, I posit that the study will make a noteworthy contribution to the existing academic literature. However, it is imperative to undertake a comparative analysis of the strengths and advantages inherent in this study. To facilitate such an assessment, I kindly propose the addition of a dedicated discussion section aimed at contrasting and highlighting the study's superiority over existing research. Furthermore, it is advisable to enhance the clarity of the abstract and conclusion by providing a more explicit exposition of the study's motivation and the results attained.

Author Response

We add our answers in a dedicated file
